# Saccharin Supplementation Inhibits Bacterial Growth and Reduces Experimental Colitis in Mice

**DOI:** 10.3390/nu12041122

**Published:** 2020-04-17

**Authors:** Annika Sünderhauf, René Pagel, Axel Künstner, Anika E. Wagner, Jan Rupp, Saleh M. Ibrahim, Stefanie Derer, Christian Sina

**Affiliations:** 1Institute of Nutritional Medicine, Molecular Gastroenterology, University Hospital Schleswig-Holstein, Campus Lübeck, 23538 Lübeck, Germany; annika.suenderhauf@uksh.de (A.S.); stefanie.derer@uksh.de (S.D.); 2Institute of Nutritional Medicine, Molecular Gastroenterology, University Hospital Schleswig-Holstein, Campus Lübeck and Institute of Anatomy, University of Lübeck, 23538 Lübeck, Germany; pagel@anat.uni-luebeck.de; 3Institute of Experimental Dermatology, Medical Systems Biology Group and Institute of Cardiogenetics, University of Lübeck, 23538 Lübeck, Germany; axel.kuenstner@uni-luebeck.de; 4Institute of Nutritional Medicine, Molecular Nutrition, University Hospital Schleswig-Holstein, Campus Lübeck, 23538 Lübeck, Germany; anika.wagner@uni-giessen.de; 5Department of Infectious Diseases and Microbiology, University of Lübeck, 23538 Lübeck, Germany; jan.rupp@uksh.de; 6Lübeck Institute of Experimental Dermatology and Center for Research on Inflammation of the Skin, University of Lübeck, 23538 Lübeck, Germany; saleh.ibrahim@uksh.de; 7Institute of Nutritional Medicine and 1st Department of Medicine, Section of Nutritional Medicine, University Hospital Schleswig-Holstein, Campus Lübeck, 23538 Lübeck, Germany

**Keywords:** saccharin, non-caloric artificial sweetener, inflammatory bowel disease, intestinal inflammation, colitis

## Abstract

Non-caloric artificial sweeteners are frequently discussed as components of the “Western diet”, negatively modulating intestinal homeostasis. Since the artificial sweetener saccharin is known to depict bacteriostatic and microbiome-modulating properties, we hypothesized oral saccharin intake to influence intestinal inflammation and aimed at delineating its effect on acute and chronic colitis activity in mice. In vitro, different bacterial strains were grown in the presence or absence of saccharin. Mice were supplemented with saccharin before or after induction of acute or chronic colitis using dextran sodium sulfate (DSS) and the extent of colitis was assessed. Ex vivo, intestinal inflammation, fecal bacterial load and composition were studied by immunohistochemistry analyses, quantitative PCR, 16 S RNA PCR or next generation sequencing in samples collected from analyzed mice. In vitro, saccharin inhibited bacterial growth in a species-dependent manner. In vivo, oral saccharin intake reduced fecal bacterial load and altered microbiome composition, while the intestinal barrier was not obviously affected. Of note, DSS-induced colitis activity was significantly improved in mice after therapeutic or prophylactic treatment with saccharin. Together, this study demonstrates that oral saccharin intake decreases intestinal bacteria count and hence encompasses the capacity to reduce acute and chronic colitis activity in mice.

## 1. Introduction

Inflammatory bowel disease (IBD) with its two main subentities, Crohn’s disease (CD) and ulcerative colitis (UC), comprises a group of chronic, relapsing-remitting, immune-mediated inflammatory disorders of the human gastrointestinal tract. Despite extensive research on IBD during the last decades, the exact etiology of both CD and UC is still unknown. Besides the impact of certain gene polymorphisms conferring the risk of developing UC [1], several experimental studies propose that the pathophysiology of UC might be additionally driven by the interplay of environmental and genetic factors, the gut microbiota and the intestinal mucosa [2,3,4,5].

One group of nutritional compounds which has been discussed to confer to the risk of UC is non-caloric artificial sweeteners (NAS) in general and saccharin (C_7_H_5_NO_3_S) specifically [6]. These synthetic compounds are hundreds of times sweeter than the table sugar sucrose with saccharin itself having a sweetness 200–700 times of table sugar [7]. The health risk of saccharin has been debated over the last decades after being found to promote urinary bladder carcinomas in mice [8]. Due to the fact that subsequent studies failed to confirm a link between saccharin and bladder cancer in humans [9], saccharin was removed from the list of potential carcinogens in the year 2000. Nowadays, saccharin is approved by the United States Food and Drug Administration (FDA) as a high-intensity sweetener, with an acceptable daily intake (ADI) of 15 milligrams per kilogram body weight per day (mg/kg bw/d) considered as safe in humans [7]. The majority of ingested saccharin is absorbed in the small intestine and eliminated unchanged via the urine while the remainder is excreted via the feces [10,11]. Of note, absorption differs between animal species depending on the stomach pH, with increased absorption found in species with a lower stomach pH, such as humans, compared with species with a higher stomach pH, including rats and mice [12]. This phenomenon will most likely result in a higher proportion of saccharin reaching the gastrointestinal tract in rodents compared to humans.

Furthermore, saccharin is proposed to bind to and signal via specific taste receptors, not only in the oral cavity but also alongside the gastrointestinal tract. Saccharin has been reported to bind the human and rodent heteromeric guanine nucleotide-binding protein (G protein) coupled sweet taste receptors T1R2/T2R3 [13,14] as well as the human bitter taste receptor T2R43 and T2R44 [15]. In the mammalian gut, T1R2 and T1R3 as well as the taste G protein α-gustducin are present on enteroendocrine L-cells of the small intestine, which control the release of peptide hormones like glucagon-like peptide-1 (GLP-1), glucagon, neuropeptide Y, peptide YY (PYY) and cholecystokinin (CCK) in response to nutrients [16,17]. Upon stimulation with sweet stimuli, GLP-1 and CCK release has been observed in the human enteroendocrine cell line STC-1 [18] and a T1R2/T1R3-dependent secretion of GLP-1 and PYY has been shown in humans [19]. However, whether saccharin ingestion also leads to hormone release via taste receptor stimulation on enteroendocrine cells in vivo remains elusive.

Saccharin is not metabolized after ingestion [10,11] and the proportion of saccharin which reaches the gastrointestinal tract is likely to come into direct contact with the intestinal microbiota. The question of how saccharin affects bacteria has been addressed both in vitro [20,21,22] and in vivo [22,23,24]. An early study from Linke et al. (1985), studying bacteria from the oral cavity [20], as well as two other studies screening bacteria mainly from the gastrointestinal tract [21,22], demonstrated a dose-dependent inhibitory effect of saccharin on bacterial growth in vitro. Furthermore, in vivo studies unraveled NAS to alter the gut microbiome in a range of experimental animal species [22,23,24]. Saccharin specifically was shown to change the rat microbiome by shifting the anaerobic/aerobic bacteria ratio, resulting in a dominance of aerobic bacteria [22]. In IBD there is growing evidence for a shift from dominance of obligate anaerobes towards facultative anaerobes and aerobes [25], but whether the observed changes are causative or are a consequence of the disease remains under debate. Daly et al. (2013) and Suez et al. (2014) revealed changes in the gut microbiome composition on the phyla and species level using next generation sequencing (NGS) techniques in swine [24] and mice [23], respectively. Interestingly, studies in different species resulted in opposing results. While Daly et al. described an over-representation of Lactobacilli in saccharin fed swine [24], Suez et al. detected a depletion of Lactobacilli, but an increase in Enterobacteriaceae and Bacteroidetes in saccharin-supplemented mice [23]. Experimental studies on the effect of saccharin-induced microbiome changes on body metabolism and intestinal health are limited to a study from Suez et al., where mice receiving NAS-supplemented drinking water developed glucose intolerance [23]. 

Data on NAS in the context of gut health in general and in humans specifically are rare and solely based on association studies. Quin et al. published a hypothesis in 2012 claiming saccharin consumption to be causative for the rising incidence of IBD cases, based on the observation that saccharin consumption in the United States significantly correlates with new IBD cases in Monroe County, New York between 1918 and 1970 [6]. As a possible mechanism for saccharin impacting the gut homeostasis, Quin previously suggested saccharin to inhibit inactivation of digestive proteases, resulting in digestion of the protective mucus layer [26]. Interestingly, neither the effect of saccharin on the course of intestinal inflammation nor the proposed mechanism of saccharin-bacteria-protease-mucus interaction have been tested experimentally to date.

Here, we aimed at investigating the impact of saccharin on bacterial growth and mucosal health as well as on intestinal inflammation in a therapeutic and a prophylactic model of DSS-induced colitis in mice.

## 2. Materials and Methods

### 2.1. Bacterial Growth Assays

Growth of following bacteria strains was tested in the presence or absence of saccharin: Firmicutes (bacilli): *Staphylococcus aureus* (ATCC 25923), *Bacillus cereus*; Proteobacteria (γ-proteobacteria): *Klebsiella pneumonia* (ATCC 183883), *Pseudomonas aeruginosa* (ATCC 27893). Bacteria were kept in sustained colonies on agar plates at 37 °C, 5% CO_2_. For growth assays, 4*10^7 bacteria/mL were stimulated with 0.5 mM, 2.5 mM and 5 mM saccharin in 33% *v*/*v* 2YT medium at a final volume of 150 µL in 96-well U-bottom plates. Optical density (OD) at 570 nm was directly measured after plate loading (time point 0) and every hour during the following 12 h to 14 h on the infinite 200 system (Tecan, Männedorf, Switzerland), kept at a stable temperature of 37 °C.

### 2.2. Animal Housing and Saccharin Supplementation 

Male C57BL/6JRj wild type (wt) mice (Janvier Labs, Le Genest-Saint-Isle, France) were maintained at the University of Lübeck under specific pathogen-free conditions at a regular 12-hour light–dark cycle with free access to food (Altromin #1324, Lage, Germany) and water. All animal experiments were approved by the ethic committee, Schleswig Holstein, Germany (V 242–7224. 122–4 (14–1/15) and V 241–45659/2016 (89–7/16)). Procedures involving animals and their care were conducted in accordance with national and international laws and regulations. Before application, saccharin (Sigma-Aldrich, St. Louis, MO, USA) endotoxin level were tested with the Pierce LAL chromogenic endotoxin quantitation kit (ThermoFisher Scientific, Waltham, MA, USA) according to protocol. Saccharin was applied via the drinking water for given periods of time in accordance with the FDA acceptable daily intake (ADI) in humans of 5 mg per kg body weight (bw). Dosage was adjusted to average mouse weight and water intake as implied by Suez et al. [27] as follows: (ADI 5 mg/kg bw/day × average mouse weight 0.03 kg)/(average water intake 2 mL/day) = 0.075 mg/mL → 0.1 mg/mL). Average water intake per cage was measured by weighing of bottles in regular intervals, followed by division via the number of mice/cage for an average water consumption/mouse. In the first round of acute colitis and both rounds of chronic colitis experiments, mice were kept in groups of 3 mice/cage, while mice were kept in groups of 4 mice/cage during the second round of acute colitis experiment. Of note, moving of cages and bottles led to water loss, resulting in higher overall calculated volumes of water consumption but were kept the same for all groups.

### 2.3. Quantification of Fecal Bacterial Load

For DNA isolation, fecal samples were collected from 6 representative mice per group selected out of at least 3 different cages per group, and DNA was extracted utilizing the QIAamp DNA stool mini kit (Qiagen, Hilden, Germany) according to the manufacturer’s instructions. Polymerase-chain reaction (PCR) was performed using the DreamTaq PCR Master Mix (ThermoFisher Scientific, Waltham, MA, USA) applying 1 µl DNA eluate and 0.5 µM of universal 16 s ribosomal RNA (16S rRNA) primer EUB338 (for: 5′-AGAGTTTGATCCTGGCTCAG-3′, rev: 5′-CTGCTGCCTCCCGTAGGAGT-3′). To capture the exponential phase, amplification was run applying 5 min primary denaturation at 95 °C, 15 cycles of 30 s at 95 °C, 30 s at 45 °C and 1 min at 72 °C and final elongation for 5 min at 72 °C. PCR products were applied on an agarose gel, band intensity was quantified using the Quantum St4 gel documentation software (Vilber Lourmat, Eberhardzell, Germany) and normalized to initial fecal weight.

### 2.4. Induction of Experimental DSS-Colitis and Assessment of Clinical Scores

Activity of colitis triggered by dextran sulfate sodium (DSS) application is known to be highly variable. It has been demonstrated to depend on the time period of oral intake, concentration, molecular weight, manufacturer and batch. However, DSS-induced colitis activity is mostly affected by the intestinal microbiome and may even vary between different rooms within one animal facility, different animal care takers or different experimental time points [28]. Hence, we experimentally determined the appropriate DSS dose before each experiment, resulting in 4% *w*/*v* applied 40 kDa DSS (TdB consultancy, Uppsala, Sweden) over five days for the first round and 2% *w*/*v* 40 kDa DSS over five days for the second round of acute experimental colitis. Acute colitis was induced in 13-weeks old male mice over the drinking water. From d3 on, mice received drinking water supplemented with saccharin and DSS or continued with DSS treatment alone (DSS-ctrl). On d5 the experiment was either stopped or mice were fed with saccharin-supplemented or normal drinking water for another two to five days. The disease activity index (DAI) was assessed daily during and post the DSS-supplementation period and was scored as follows: body weight: 0, no weight loss or consistent weight; 1, 1% to 5% weight loss; 2, 5% to 10% weight loss; 3, 10% to 15% weight loss; 4, 15% to 20% weight loss; rectal bleeding: 0, no blood; 2, positive; and 4, gross blood; and stool consistency: 0, solid; 2, soft; 4, diarrhea. The modified disease activity index only included the parameters rectal bleeding and stool consistency. Animals that lost more than 20% of their initial weight were euthanized and excluded from further analysis. Mice were sacrificed at indicated time points d5, d7 and d10 post start of DSS-supplementation and blood, stool and tissue samples were taken for further analysis. 

For chronic colitis experiments, 12-weeks-old male mice which had been receiving saccharin for five weeks or were left untreated (ctrl) were exposed to three cycles of five days 2% *w*/*v* 40 kDa DSS (TdB consultancy, Uppsala, Sweden) via the drinking water, interrupted by five days of normal drinking water each. The DAI was assessed every other day as described above and all mice were sampled at day 30 post start of chronic colitis. 

### 2.5. Colonoscopy and Histology

For endoscopic examination, mice were sedated by intraperitoneal injection of 100 mg/kg ketamine (CGM Medistar, Hannover, Germany) and 10 mg/kg xylazin (Bayer, Leverkusen, Germany). High-resolution mouse endoscopy (AIDA compact NEO, Karl Storz, Tuttlingen, Germany) was used to macroscopically examine the constitution of the colonic mucosa. Preparation of colonic sections and haematoxylin and eosin (HE) staining was performed according to standard protocols as described earlier [29]. For periodic-acid Schiff (PAS)/alcian blue staining, colonic biopsies were fixed using Carnoy’s solution without prior flushing with PBS. The histology score for HE-stained colonic sections was applied in accordance with the scoring system from Siegmund et al., combining leucocyte infiltration and tissue damage [30]. Leukocyte infiltration was scored as follows: 0: occasional leucocytes in the lamina propria, 2: confluence of leucocytes extending into the submucosa, 3: transmural extension of the infiltrate. Tissue damage was categorized accordingly: 0: no mucosal damage, 1: lymphoepithelial lesions, 2: surface mucosal erosion or focal ulceration, 3: extensive mucosal damage and extension into deeper structures of the bowel wall. The latter resulted in a histological score, ranging from 0 (no abnormality) to 6 (extensive cell infiltration and tissue damage). Tissue samples were microscopically scored using the light microscope AxioPhot (Zeiss, Oberkochen, Germany).

### 2.6. RNA Extraction and Quantitative Real-Time PCR

RNA was extracted and transcribed to cDNA as described earlier [31]. Quantitative real-time polymerase-chain reaction (qPCR) was performed using the PerfeCTa SYBR Green SuperMix (Quantabio, Beverly, MA, USA) and specific oligonucleotides on a StepOnePlus System (Applied Biosystems, Waltham, MA, USA). Amplification was run, applying 10 min of preincubation at 95 °C followed by 40 cycles of 15 s at 95 °C, 30 s annealing at a primer-dependent temperature of 55 °C to 60 °C and 30 s elongation at 72 °C. Expression levels were normalized to ß-actin applying the dCt algorithm. The following oligonucleotides were used: *ß-actin* for: 5′-GATGCTCCCCGGGCTGTATT-3′, rev: 5′-GGGGTACTTCAGGGTCAGGA-3′; *kc* for: 5′-GCTGGGATTCACCTCAAGAA-3′, rev: 5′-TGGGGACACCTTTTAGCATC-3′; *icam1* for: 5′-GTGATGCTCAGGTATCCATC-3′, rev: 5′-GTCCACTCTCGAGCTCATC-3′; *t1r2* for: 5′-GACACTCCATTTGCTGTTTC-3′, rev 5′-GGCAGGAAGTCATCTATCTG-3′; *t1r3* for 5´-GTCACCATGAAATCCAGTCT-3´, rev 5´-GCCAGACTAGAAAAGATGCT-3´; *pyy* for: 5′-CTGCTAATCCTGCTCGCCTG-3′, rev 5′-CCTGAAGGGGAGGTTCTCG-3′; *cck* for: 5′-CTTAAGAACCTGCAGAGCCTGG-3′, rev: 5′-TTTCCTCATTCCACCTCCTCC. 

### 2.7. Murine Cxcl1/Kc Enzyme Linked Immunosorbent Assay (ELISA)

For serum preparation, whole blood, which was collected directly from the heart of sedated mice, was allowed to clot and serum was separated from plasma by centrifugation. Mouse cxcl1/kc DuoSet ELISA (R&D Systems, Minneapolis, MN, USA) was performed to analyze serum kc level according to manufacturer’s instructions. Serum samples were utilized in a 1:10 dilution and final OD was measured at 450 nm against a reference wavelength of 540 nm on a SpectraMax iD3 plate reader (Molecular Devices, San Jose, CA, USA).

### 2.8. Intestinal Microbiota Analysis

Fecal samples were collected and directly frozen in liquid nitrogen from 48 animals (16 cages, 3 mice per cage) before treatment, and from the according 24 saccharin-supplemented and 24 control animals (8 cages each, 3 mice per cage each) after the five weeks supplementation period. DNA extraction from frozen fecal samples with the PowerSoil DNA Isolation kit (Qiagen, Hilden, Germany), followed by PCR amplification of 16S rRNA hypervariable regions V1 and V2, library preparation and final sequencing on the Illumina MiSeq system (paired-end reads, 2 × 300 base pairs; San Diego, CA, USA) were performed as described earlier [3]. For data processing, overlapping raw sequencing reads were merged into contigs USEARCH version 8.0.1623 [32] (fastq_merge_pairs command; maximum number of mismatches: 8), quality filtered (fastq_filter; expected number of errors set to 0.25) and chimeras were removed with the RDP Gold database version 9 [33] as reference database (uchime_ref). To perform taxonomic classification the RDP version 14 and MOTHUR (version 1.34.4) [34] with 80% bootstrap support was used. Only sequences were kept that were confirmed as bacterial origin. Sequences were aligned to the 16S rRNA V1-V2 region with SILVA (version 123) database as reference. For each sample, 10,000 sequences were randomly chosen to normalize for different sequence numbers between the individuals. Operational taxonomic units (OTUs) were assigned at 97% similarity threshold and the representative sequence for each OTU was chosen using a distance method as implemented in MOTHUR. A phylogenetic tree was constructed by using FASTTREE version 2.1.4 [35] with a generalized time-reversible (GTR) substitution model, and the gamma option to rescale branch length. The final tree was rooted using midpoint rooting. Unifrac distance (beta diversity) was estimated using phyloseq package version 1.26.0 and analyzed performing non-parametric matrix-based analysis of variance by using adonis (vegan package for R; 999 permutations to compute significant values for each factor). Constraint analysis of principle coordinates were performed using capscale (vegan package version 2.5–3) and significance of constraints was assessed with 1000 permutations for each factor using an ANOVA-like permutation test. All statistical analyses were performed applying R version 3.3.0. Heatmaps from indicator species analysis were generated using the Morpheus software [36]. Significance was assessed using Mann–Whitney *U* test or Kruskal–Wallis with post-hoc testing (Dunn-test with Benjamini–Hochberg correction for multiple testing) depending on the data distribution and number of groups. 

### 2.9. Protein Extraction, SDS-PAGE and Immunoblotting

For extraction of proteins from fecal samples, 50 mg feces was resuspended in 1 mL 0.5% *v*/*v* Tween, 0.05% *w*/*v* sodium azide in PBS and centrifuged at 12.000× *g* for 15 min at 4 °C. Supernatant was taken off, protease inhibitor cocktail (Sigma-Aldrich, St. Louis, MO, USA) was added, samples were centrifuged as above and supernatant was taken for further analysis. Western blotting (WB) was performed according to standard protocols. Briefly, 9 µg protein lysates were separated by SDS-PAGE (Criterion TGX Precast Gel 4–15%, Bio-Rad, München, Germany), transferred to a PVDF membrane using the Trans-Blot semi-dry System (Bio-Rad, Hercules, CA, USA). Membranes were incubated with an antibody against the murine IgA alpha-chain (Abnova, Taipei, Taiwan) and the according horseradish peroxidase-conjugated secondary antibody.

### 2.10. Statistical Analysis

Results are expressed as mean ± standard error mean (SEM), if not stated otherwise. Statistical analysis was performed using the GraphPad Prism version 6 for Windows (GraphPad Software, San Diego California USA). Outliers were identified by applying the Prism ROUT method; Q = 1%. Normal distribution of data was tested using the D’Agostino–Pearson test. Statistical differences of two groups were analyzed by unpaired t-test (normally distributed data) or the Mann–Whitney U-test (not-normally distributed data). Grouped analysis involving testing of a parameter between ctrl and saccharin groups vs time was analyzed via two-way ANOVA, comparing means of each time point followed by Sidak testing for multiple comparisons. For grouped analysis including testing for differences in ctrl and saccharin groups with different measurement parameters, multiple t-tests were used correcting for multiple comparisons using the Holm-Sidak method, α = 5%. *p* ≤ 0.05 = *; *p* ≤ 0.01 = **; *p* ≤ 0.001 = ***; *p* ≤ 0.0001 = ****.

## 3. Results

### 3.1. Saccharin Reduces Bacterial Growth In Vitro and In Vivo 

Alterations in intestinal bacterial composition is often observed in IBD and intestinal bacterial overgrowth confers the risk of intestinal inflammation [5,37]. Therefore, we first studied the direct effect of saccharin on bacterial growth from exemplary bacterial species, with 5 mM saccharin exhibiting bacteriostatic but not bactericidal effects on *Staphylococcus aureus* (Firmicute), *Klebsiella pneumonia* and *Pseudomonas aeruginosa* (both Proteobacteria) (Figure 1a,b). Contrastingly, stimulation of *Bacillus cereus* (Firmicute) with 5 mM saccharin resulted in a complete abrogation of bacterial growth, supporting the idea that saccharin has different inhibitory efficacies on different bacterial species. To validate bacteriostatic effects of saccharin in vivo, mice were supplemented with saccharin via the drinking water over a period of 35 days or were left untreated. In line with in vitro data, saccharin-supplemented mice depicted reduced levels of bacterial 16S rRNA in fecal samples compared to controls (Figure 1c). 

### 3.2. Saccharin Supplementation Does Not Substantially Alter the Intestinal Barrier

Bacterial overgrowth contributes to intestinal inflammation and here we show that saccharin confines bacterial growth. We therefore tested for weight gain, mucosal damage and molecular markers of inflammation in C57BL/6JRj wt mice, which received 0.1 mg/mL of endotoxin-free saccharin (Figure 2a) via the drinking water for seven days. Here, neither water consumption (Figure 2b) nor weight development (Figure 2c) or final colon length (Figure 2d) were significantly different between saccharin-supplemented and control mice. Furthermore, there were no signs of inflammation in HE-stained colonic sections, no reduction of goblet cells or mucus production in PAS-stained colonic tissue and no visible mucosal damage in colonoscopic imaging (Figure 2e). Next, we tested for mRNA and protein expression of KC, a neutrophil recruiting chemokine, which is induced by various inflammatory stimuli in immune as well as epithelial cells [38,39], and mRNA expression of intercellular adhesion molecule (icam-1), a major regulator of inflammatory signaling, which is upregulated upon nuclear factor kappa B activation [40]. Surprisingly, colonic *icam-1* mRNA levels were significantly decreased after five days of saccharin supplementation compared to controls, while ileal *icam-1* mRNA expression showed a trend towards reduction (*p* ≤ 0.08) after three days of saccharin intake compared to controls (Figure 2f). Of note, saccharin-supplemented mice in comparison to control mice further displayed diminished serum protein level of KC (Figure 2g) and a trend towards reduced ileal *kc* mRNA level (*p* ≤ 0.08) (Figure 2f). These data show that in contrast to the primary hypothesis, saccharin did not induce intestinal inflammation but even lowered expression of inflammatory markers in supplemented mice. Since saccharin could have an effect on the intestinal epithelium not only via alterations of the microbiota but also directly via taste-receptor signaling, we analyzed taste receptors and the according neurotransmitter expression in the intestine of saccharin-treated versus ctrl mice. Notably, while t1r3 mRNA expression levels were similar throughout the intestinal tract of untreated C57BL/6J wt mice, t1r2 depicted very low expression levels and was solely detectable in duodenal and jejunal tissue sections (Appendix A). We therefore used biopsies from the proximal jejunum of seven day saccharin-treated and ctrl animals to test for mRNA expression level of t1r2 and t1r3. No differences were observed either in taste receptor expression (Appendix A) nor in taste-receptor associated neurotransmitter cck or pyy mRNA level (Appendix A).

### 3.3. Saccharin Treatment after Induction of Experimental Colitis Improves Intestinal Inflammation in Mice

After the surprising finding of an improvement of gut inflammatory parameters in saccharin-receiving mice, an acute experimental colitis was induced in C57BL/6J wt mice by supplementing the drinking water with DSS for five days. To test the effect of saccharin administration in a therapeutic model, acute colitis was first induced by oral DSS application, followed by saccharin supplementation of according groups from day three onwards, when colitis was established but not too far progressed (Figure 3a). Of note, water consumption was not different between groups (Figure 3b). Survival of mice according to abrogation criteria until day ten post start of colitis was similar between saccharin-treated and untreated groups (Figure 3c). Nevertheless, the modified DAI, combining rectal bleeding and stool consistency, was significantly decreased in saccharin-supplemented mice five and six days post start of DSS treatment (Figure 3d). This was also reflected by a significantly decreased area under the curve (AUC) of the modified DAI in DSS-saccharin treated mice compared to DSS-control mice (Figure 3e). Loss of body weight loss, was not different between analyzed groups (Figure 3f). In line, analysis of molecular markers of inflammation revealed significantly lower *icam-1* mRNA level in the ileum and a trend towards less *kc* mRNA transcript in the colon of saccharin mice compared to ctrl mice five days post start of DSS-treatment (Figure 3g). The KC serum level and the colonic histology score were significantly decreased in the DSS-saccharin group sampled on day seven compared to controls (Figure 3h–j), further supporting the idea of a protective effect of saccharin in DSS-induced experimental colitis.

### 3.4. Saccharin Is Protective in a Prophylactic Mouse Model of Experimental Colitis

After having shown that saccharin has protective effects, when applied therapeutically after induction of experimental DSS-colitis, we were wondering if this would also be the case when employing a prophylactic model of five weeks pre-feeding with saccharin before colitis induction (Figure 4a). In Figure 1c we have already shown, in a subset of representative animals, that a five-week application period of saccharin leads to a reduction of fecal bacterial load in saccharin-supplemented mice compared to controls. Previous studies have shown saccharin to induce changes in microbiome composition in vivo [23,24]. In line with this, we observed a significant influence of saccharin on beta-diversity (UniFrac ADONIS (1000 permutations), *p =* 0.014; Figure 4b). Alpha-diversity was not different between groups (Mann–Whitney *U* test Shannon *p =* 0.4429 and Chao1 *p =* 0.9919), Appendix A), while mean phyla abundance and indicator species analysis revealed a shift towards an increase in the Bacteroidetes and Proteobacteria phylum and a decrease in the Firmicutes phylum (Appendix A). Alterations in microbiome composition might be explained by distinct bacteriostatic activity of saccharin on different bacterial species. In line with comparatively strong inhibition of Firmicute *Bacillus cereus* (Figure 1a) in in vitro experiments, relative abundancy of the phylum Firmicutes was also reduced in the microbiome analysis of fecal samples of saccharin-supplemented mice compared to controls. Vice versa, a decrease in Bacteroidetes and an increase in Firmicutes is associated with obesity and metabolic syndromes in humans [41]. Water consumption and weight gain were statistically not different between saccharin-supplemented and ctrl mice throughout the five weeks pre-feeding period (Appendix A).

After induction of microbiome alterations, all mice were set back to normal drinking water and an experimental colitis was induced via DSS in previously saccharin-treated or control mice (Figure 4a). Of note, DSS-water consumption was not different between groups (Figure 4c), but was overall higher compared to water consumption measured during acute colitis (Figure 3b). This is due to the fact that water consumption in the acute setup was measured in mice, which had already received 3 days of DSS-treatment and had consumed less water due to a poor overall wellbeing. In line with the results from the acute colitis, previously saccharin-supplemented mice had significantly less weight loss compared to controls (Figure 4f). Further, DAI levels at day 10, 13 and 20 after colitis induction were significantly lower in saccharin pre-treated mice (Figure 4d), which was also reflected by a significantly lower AUC of the DAI of saccharin pre-treated mice compared to ctrl mice (Figure 4e). The differences between groups did not endure until the end of the experimental period. Most likely the positive effects of a saccharin-induced microbiome disappeared at later stages of the chronic colitis model (d22–d30), since saccharin supplementation was stopped at the beginning of DSS treatment (d0). Therefore, we investigated IgA level in fecal samples from day 30 as a marker for B-cell differentiation induced by recent inflammatory activity. Western blotting experiments revealed significantly lower levels of dimeric IgA in mice which underwent five weeks of saccharin-supplementation followed by chronic DSS-colitis, compared to controls (Figure 4g). Taken together, data suggest a saccharin-induced microbiome, including the characteristics of a reduced bacterial load and a relative decrease in the Firmicutes phylum, to have anti-inflammatory properties in experimental DSS-colitis.

## 4. Discussion

Increased consumption of non-caloric artificial sweeteners has been claimed to be associated with the rising incidence of inflammatory bowel disease [6]. Nevertheless, a recent review found no clinical evidence of an inflammatory effect on the gut caused by non-caloric sweeteners in general [42]. Here, we found saccharin consumption in mice to even protect from intestinal inflammation. In a therapeutic model of saccharin supplementation in mice with an established acute DSS-induced colitis, we were able to observe a reduction of disease activity on the molecular and macroscopic level. Based on these findings, we hypothesize that this effect was most likely due to a species-specific inhibitory effect of saccharin on bacterial growth and therefore a change in bacterial composition in saccharin-supplemented mice. A prophylactic model of saccharin supplementation in mice revealed saccharin intake to alter intestinal microbiome composition, characterized by a decrease in bacterial load and an increase in relative abundancy of Bacteroidetes, thereby potentially reducing chronic colitis activity in early stages of experimental colitis.

Saccharin has been published to cause liver inflammation [43] and glucose intolerance [23] both by altering the gut microbiome composition. Here, we confirm a shift of the intestinal bacterial population after saccharin treatment. In line with data from Suez et al. (2014) [23], we detected an increased abundance of Bacteroidetes and a decreased abundance of Firmicutes in saccharin-fed mice, but no changes in alpha-diversity.

In line with previous findings [21,22], this study shows oral saccharin intake to alter microbiome composition as well as to reduce the fecal bacterial load. Bacteria which come in close contact with epithelial cells can cause severe inflammatory reactions and small intestinal bacterial overgrowth (SIBO) is often observed in patients diagnosed with CD or UC [37]. In the treatment of pouchitis and SIBO, antibiotics are frequently used to lower the bacterial burden and to thereby overcome acute epithelial inflammation. The present study reveals the bacteriostatic effects of saccharin intake in mice leading to the hypothesis that saccharin consumption may serve as a potential alternative to antibiotic treatment. Nevertheless, murine DSS-colitis as a model of intestinal integrity loss only mimics the epithelial damage observed in IBD, is restricted in its transferability towards the human system and saccharin’s exact mode of action is still unknown. Further, due to increased absorption rates of saccharin in humans compared to mice [12], most likely a lower proportion of ingested saccharin will reach the intestinal tract in humans. Therefore, higher doses of saccharin are potentially required to induce similar effects on the intestinal microbiota compared to mice models, also increasing the risk of adverse effects.

Saccharin might interact directly with bacteria via sensory molecules on the bacterial surface or effects on bacterial growth might be secondary. In solution, saccharin depicts acidic properties and intestinal luminal pH level of 5.5 have been demonstrated to favour butyrate-producing bacteria in human microbiome studies [44]. Interestingly, butyrate especially depicts protective effects in experimental colitis and human IBD [45]. In contrast to this hypothesis, Suez et al. (2014) [23], found saccharin-supplemented mice to have increased amounts of fecal acetate and propionate compared to controls, while no information is available on butyrate levels after saccharin feeding.

A second hypothesis concerning saccharin’s mode of action is the binding to and activation of the taste receptor heterodimer T1R2/T1R3 which has been shown to be expressed in the human intestine [16]. Stimulation of taste receptor expressing cell lines with saccharin has been reported to result in release of peptide hormones CCK and GLP-1 [18]. Both of these have been described to exhibit anti-inflammatory properties in the context of type 2 diabetes [46,47], but were also proposed to have anti-inflammatory properties in the context of experimental colitis [48]. Nevertheless, we only found very low expression levels of T1R2 and T1R3 in mice and did not see an effect of saccharin treatment on taste receptor associated neurotransmitter expression in the gut.

This study suggests that the positive effects of saccharin on intestinal inflammation might outweigh potential negative effects in specific clinical situations of bacterial overgrowth such as SIBO and pouchitis. However, several functional and toxicity studies are needed to understand the specific effects of NAS on the microbiome, the intestinal mucosa, and its health effects.

## Figures and Tables

**Figure 1 nutrients-12-01122-f001:**
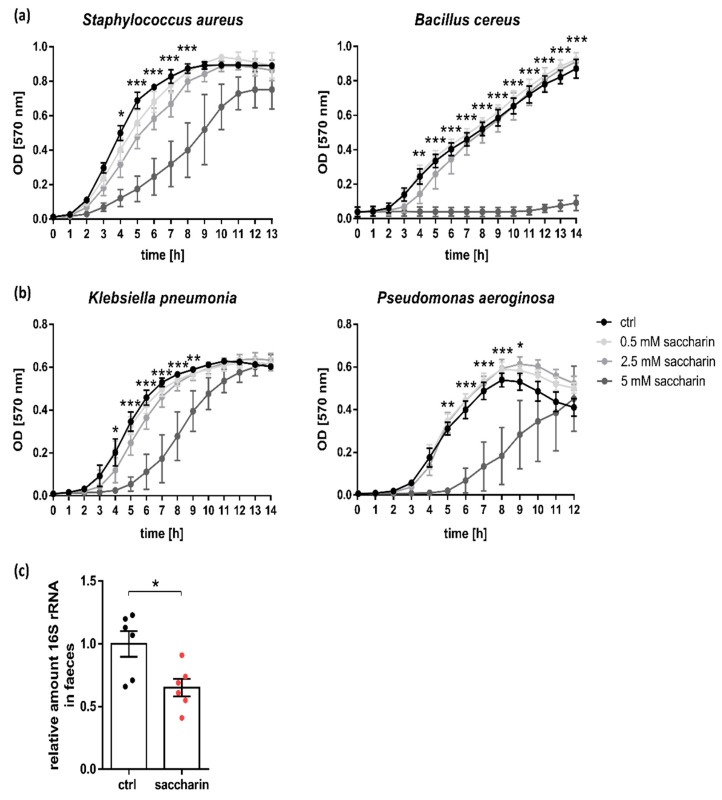
Saccharin inhibits bacterial growth in vitro and in vivo. Bacterial growth curves of (**a**) Firmicutes and (**b**) Proteobacteria in the presence or absence of different saccharin concentrations; *n* = 3. Statistical significance is shown for 5 mM saccharin versus ctrl analyzed by two-way ANOVA followed by Tukey testing for multiple comparisons. (**c**) Bacterial load in the feces after 35 days of saccharin supplementation or ctrl feeding was quantified by PCR and is shown as relative amount of 16S rRNA to ctrl; *n* = 6. Values are shown as mean ± SEM. * *p* ≤ 0.05; ** *p* ≤ 0.01; *** *p* ≤ 0.001.

**Figure 2 nutrients-12-01122-f002:**
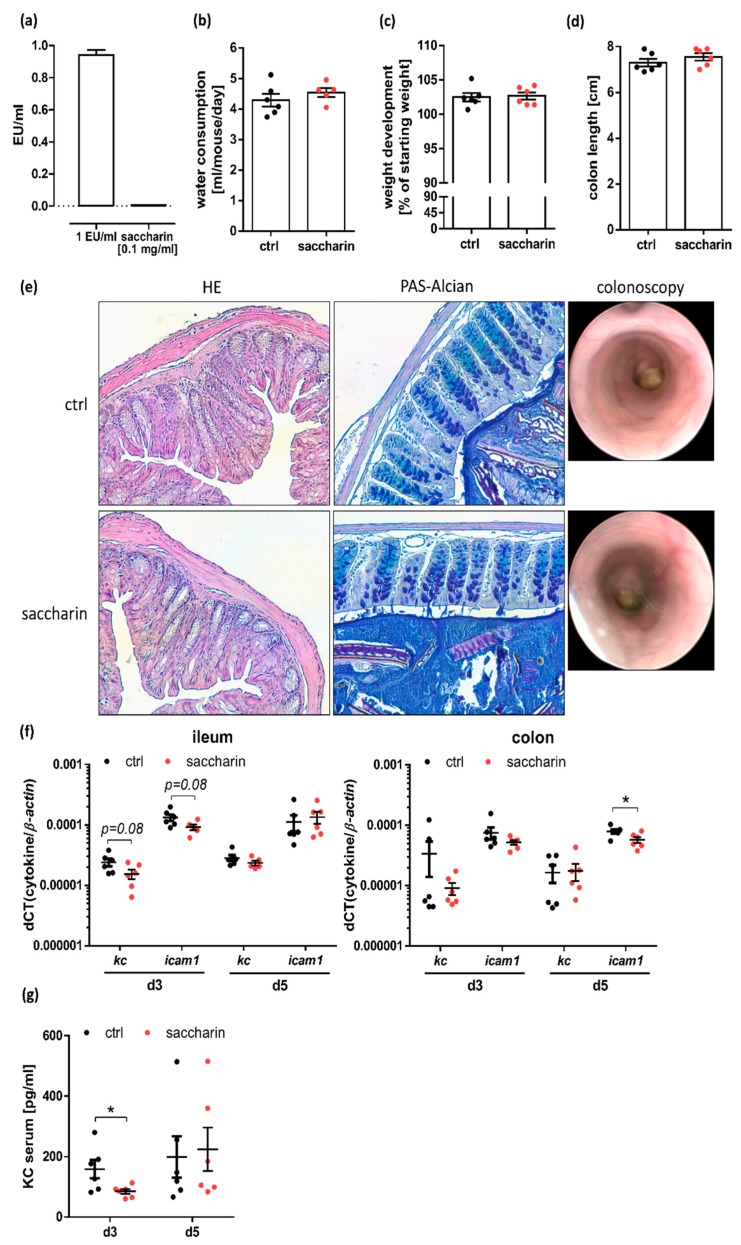
Short-term saccharin supplementation does not lead to intestinal barrier dysfunction but even lowers expression of inflammatory marker in mice. (**a**) Endotoxin units (EU) of applied saccharin compared to positive control of 1 EU/mL. (**b**) Water consumption, (**c**) weight development and (**d**) colon length of mice which were supplemented for seven days with 0.1 mg/mL saccharin via the drinking water or left untreated; *n* = 6. (**e**) Representative HE and PAS-Alcian stainings of colonic tissue samples as well as representative images from colonoscopy obtained from mice after seven days of saccharin feeding. (**f**) Analysis of inflammatory marker expression in ileal and colonic tissue was perdormed by rt-qPCR and normalized to *ß-actin*. (**g**) KC level in serum sampled after three or five days of saccharin supplementation was measured by sandwich ELISA; *n* = 6. Values are shown as mean ± SEM. * *p* ≤ 0.05.

**Figure 3 nutrients-12-01122-f003:**
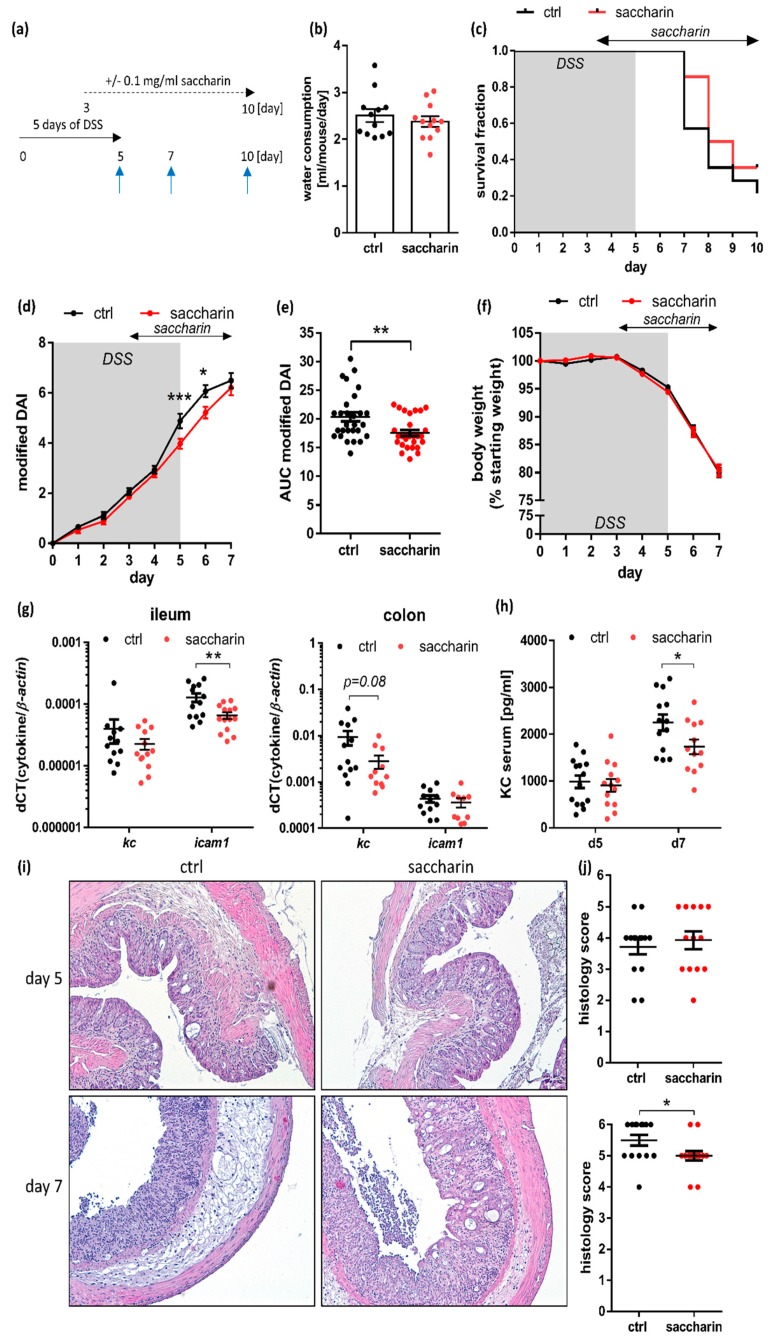
Saccharin treatment of mice after induction of an acute DSS-colitis improves intestinal inflammation. (**a**) Mice were supplemented for 3 days with 2% or 4% *w*/*v* DSS via the drinking water, followed by two days of supplementation with DSS ± saccharin and a maximum of five days of ± saccharin treatment only. Blue arrows indicate days of sampling. (**b**) Water consumption of mice, which had already received three days of acute DSS treatment and were then supplemented with DSS-saccharin or DSS-ctrl. (**c**) Survival of all animals planned to be sampled on day ten, taking the abrogation criteria into account; *n* = 14 from two independent experiments. (**d**) The modified DAI and (**f**) development of body weight is shown for all mice included in the experiment until day 5 (*n* = 42 from two independent experiments), and all mice which were sampled on day 7 and day 10 post start until day 7 (*n* = 28 from two independent experiments). The latter were used to calculate in (**e**) the AUC of the modified DAI; *n* = 28 from two independent experiments. (**g**) Expression levels of inflammatory cytokines measured via qPCR experiments from ileal and colonic tissue sampled on day 5 were normalized to *ß-actin*, and (**h**) KC levels were measured via ELISA in serum samples; *n* = 11 to 14 each from two independent experiments. (**i**) Representative HE stainings and (**j**) histology scoring from DSS-saccharin and DSS-ctrl mice sampled on day 5 and day 7 post start of colitis induction; *n* = 14 animals per group per time point. Values are shown as (**c**) fraction ± SEM or (**b**) and (**d**) to (**h**) and (j) mean ± SEM. * *p* ≤ 0.05; ** *p* ≤ 0.01; *** *p* ≤ 0.001.

**Figure 4 nutrients-12-01122-f004:**
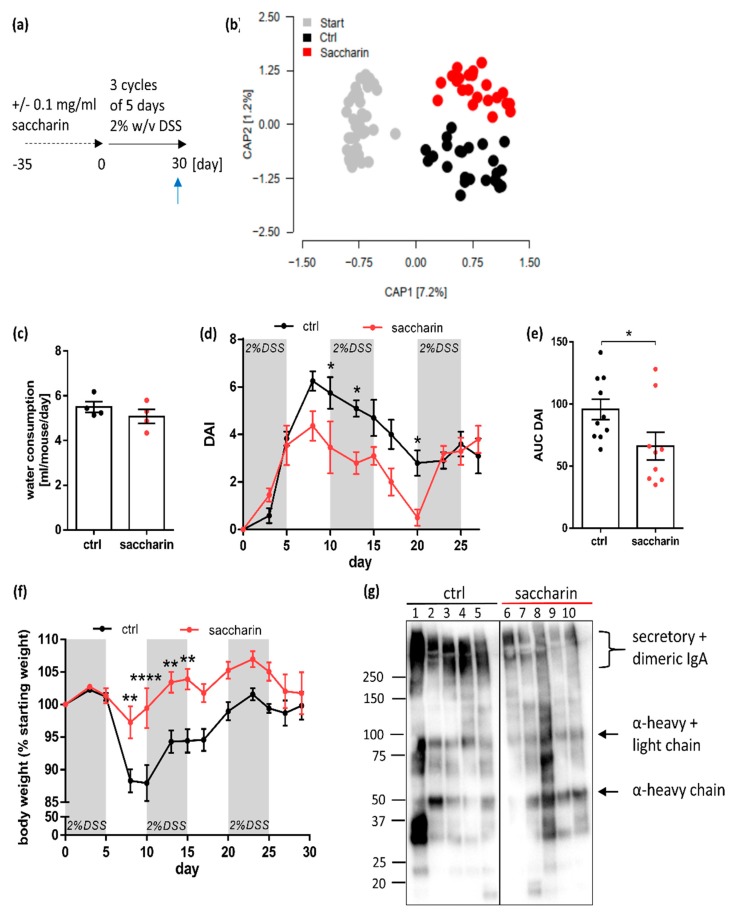
Five weeks of saccharin pre-feeding has a protective influence on chronic DSS-colitis in mice. (**a**) Drinking water of mice was supplemented with 0.1 mg/mL saccharin or left untreated for a period of five weeks, after which saccharin treatment was stopped and chronic colitis was induced in 3 cycles of 2% DSS-treatment via the drinking water. The blue arrow indicates the day of organ sampling. (**b**) Beta-diversity of fecal microbiome after five weeks of supplementation shown as unifrac-distance. Start: *n* = 48, ctrl and saccharin *n* = 24. (**c**) Water consumption during the first DSS-cycle of chronic colitis treatment of previously saccharin or untreated mice; *n* = 4 of two different experiments. Degree of intestinal inflammation is shown in (**d**) as DAI over time, in (**e**) as AUC of the DAI and in (**f**) as body weight loss over time. (**d**) and (**f**) *n* = 11–12 and (**e**) *n* = 9–10 from two independent experiments. (**g**) Denaturing, non-reducing WB of fecal proteins against IgA α-chain from DSS-ctrl or DSS-saccharin mice. Representative WB from two different experimental rounds. In (**d**) and (**f**) all mice were included until time of death while for (**e**) and (**g**) mice which were taken out at earlier time points due to abrogation criteria were excluded. Values are shown as mean ± SEM. * *p* ≤ 0.05; ** *p* ≤ 0.01; **** *p* ≤ 0.0001.

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
