# Peer review of "Saccharin Supplementation Inhibits Bacterial Growth and Reduces Experimental Colitis in Mice"

_nutrients, 2020, doi:10.3390/nu12041122_

Round 1

Reviewer 1 Report

Sünderhauf et al. investigated the effect of saccharin supplementation in the bacterial growth and its relevance in two murine models of colitis. The authors performed both in vitro and in vivo experiments with mice. The topic of this manuscript seems to be quite relevant for IBD pathology since it is still not well elucidated the effect of saccharin in gut homeostasis and in some pathologies which affect the intestine such as Inflammatory Bowel Disease. Nevertheless, this reviewer has found crucial mistakes in the experiments performed and the conclusions that authors claimed in the manuscript cannot be deduced from their results. For this reviewer, there are major concerns that must be addressed.

First of all, authors must clarify all the protocols performed with mice (the whole manuscript, except the first in vitro experiment, has been performed with mice). In both material and methods section and in figure legend of figure 3, authors described that acute colitis was performed with 4% or 2% of DSS. However, in the figure it is not clear the % of DSS used. Why authors mentioned two different percentages and do not show 2 different groups of mice with 2 and 4% of DSS? This must be clarified.

In figure 1, authors only showed the amount of 16S rRNA of mice supplemented 35 days with saccharin. It would be clearer if authors show after this result the microbiome analysis included in Supplementary Figure 2 if the mice were the same. Do these results reproduce their in vitro observations? I also do not understand why the n in figure 1c is 6 per group, while in supplementary figure 2 is 12 and 24. Which is the reason of such a big difference in the n?

In figure 2 authors state that saccharin reduces the expression of inflammatory markers. They mention that the expression of kc/cxcl1 and icam1 was reduced by saccharin but this reduction is not significant at day 3 and at day 5 there are no differences. Therefore, if authors conclude that saccharin reduces the expression of inflammatory markers, they should measure the mRNA and protein expression of more classical inflammatory markers such as TNFα, il1β, COX-2, iNOS, etc. They should also measure the MPO activity.

In figure 3, why did authors start to give saccharin at day 3? Authors must treat the mice with saccharin at the beginning or at the end of DSS treatment but not just in the middle without any explanation. In addition, authors have measured all the parameters at day 5, but the experiment finished at day 10. Authors must analyse the mice at day 10. Authors should include the graph of weight loss and not only show the AUC DAI. Here the same, why the number of mice per group is so different between Figure 1d and the gene expression of kc and icam1? Authors must show histological pictures of those mice in order to show the inflammation caused by DSS. Moreover, they must measure the mRNA and protein expression of more inflammatory markers such as TNFα, il1β, COX-2, iNOS, etc. and also the MPO activity.

In figure 4, there are not histological pictures of the mice. Authors must analyze also in these mice the expression of proinflammatory markers. The WesternBlot shown in Figure 4g is really dirty with a lot of bands. In addition, the band pointed with the arrow is not well defined. Authors must improve the protocol of the western in order to show a clearer westernblot.

In the discussion authors must discuss whether the “reduction” of the inflammation is a cause or consequence of the bacteria alteration due to the saccharin. In my opinion, authors must strengthen the “reduction” of the inflammation due to the saccharin administration.

Minor points:

  • check sentences in line 376 and 381-382
  • Line 158 there is not Figure 5a
  • Revise Figure 3. There are two d repeated
  • Check Figure legend of Figure 4 (it is not Figure 6)

Reviewer 2 Report

This manuscript discusses the beneficial role of saccharin in reducing bacterial growth in vitro and in vivo, and preventing inflammation in mice, contradicting previous studies showing adverse effects of saccharin on inflammation and microbial composition. Several mechanisms have been proposed to explain this association. The studies are well designed and currently of particular interest. I have a few comments below:

Title could be improved: maybe add ‘improves microbiome composition or reduces inflammation’?                                                                              Also, I would keep it simple. Use ‘reduce’ as opposed to ‘confine’

Abstract needs to be more clearly written. It does not clearly summarise the aim and methodology of the study. Mention aim and design of both in vitro and in vivo studies, add results and end with a short conclusion.

Line 84: It would be good to explain to the readers whether a dominance of aerobic bacteria would have beneficial or adverse effects on health.

Line 85: same comment as above.

Lines 102-104: Omit results from Introduction. The latter should end with the aim of the study.

Line 115: This paragraph seems to be out of context. There is no indication of the use of bacterial load in the faeces. Please clarify.

Line 241: My advice would be to use a different word than confine.

Line 377: Please specify what you mean by microbial alterations.

Line 401: overweigh?

Line 403: ‘However, several functional and toxicity studies are needed to understand specific effects of NAS on the microbiome, the intestinal mucosa and health effects’.

Round 2

Reviewer 1 Report

The authors have addressed most of my concerns.

Author Response

We are happy that we were able to adress the concerns of the reviewer.